# Association between self-administrated prophylactics and SARS-CoV-2 infection among traditional market vendors from the Central Highlands of Peru: A nested case-control study

Daniel A. Andrade[1], Ana Ho-Palma[2], Cesar A. Valdivia-Carrera[3,4], Astrid Munguia[4], Christine Leyns[5,6], Javier Guitian[7]*, Eloy Gonzales-Gustavson[1,3,4]

1 Center of Global Health, Universidad Peruana Cayetano Heredia, Lima, Peru, 2 Department of Human Medicine, School of Human Medicine, Universidad Nacional del Centro del Perú, Junin, Peru, 3 Department of Animal and Public Health, School of Veterinary Medicine, Universidad Nacional Mayor de San Marcos, Lima, Peru, 4 Tropical and Highlands Veterinary Research Institute, Universidad Nacional Mayor de San Marcos, Junin, Peru, 5 Research Institute of Social Sciences (INCISO), Faculty of Social Sciences, Universidad Mayor de San Simon, Cochabamba, Bolivia, 6 Department of Public Health and Primary Care, Faculty of Medicine and Health Sciences, Ghent University, Ghent, Belgium, 7 Veterinary Epidemiology, Economics and Public Health Group, Department of Pathobiology and Population Sciences, The Royal Veterinary College, Hertfordshire, United Kingdom

* jguitian@rvc.ac.uk

## Abstract

Although COVID-19 is no longer a public health emergency of international concern, understanding behaviours such as self-medication remains relevant for informing future outbreak responses and improving public health preparedness. Despite its widespread use during the pandemic, research on medications preventing SARS-CoV-2 infection in healthy individuals is scarce. We investigated the association between self-administered prophylactics and SARS-CoV-2 infection during the third wave of the pandemic in Peru. A nested case-control study was carried out in a cohort of traditional market vendors in the Peruvian Central Highlands, enrolled in a health program. Cases (positive SARS-CoV-2 diagnosis) were matched with controls (negative) by age, sex, and market of origin. Conditional logistic regression models were fitted to evaluate the association between self-administered prophylactics and SARS-CoV-2 infection. As a result, 73 cases were matched with 176 controls. Acetylsalicylic acid consumption increased SARS-CoV-2 infection odds (adjusted Odds Ratio 2.34; 95% Confidence Interval 1.17–4.66). Conversely, vitamin C consumption reduced infection odds (adjusted Odds Ratio 0.44; 95% Confidence Interval 0.23–0.87). Finally, not having the COVID-19 booster increased infection odds (adjusted Odds Ratio 3.38; 95% Confidence Interval 1.43–7.95). In conclusion, our findings suggest that acetylsalicylic acid consumption increased the odds of SARS-CoV-2 infection, whereas vitamin C consumption decreased the infection odds during

**Data availability statement:** All relevant data are within the manuscript and its Supporting Information files.

**Funding:** This study was supported by the Medical Research Council and a National Institute for Health and Care Research (NIHR) Global Effort on COVID-19 (GECO) Health Research award 2020 [Award number MR/V028561/1]. DA and EG received partial support from the Fogarty International Center/NIH [Training grant number D43TW00114]. DA also received partial support from the US Centers for Disease Control and Prevention through a cooperative agreement [Award number 5U01GH00266].

**Competing interests:** The authors have declared that no competing interests exist.

the third epidemic wave in Peru. Further research on the use of these medications is needed to establish a robust causal relationship with SARS-CoV-2 infection.

## Introduction

Since the onset of the coronavirus disease 19 (COVID-19) pandemic, severe acute respiratory syndrome coronavirus 2 (SARS-CoV-2) infection has caused high world-wide rates of morbidity and mortality [1], with Peru as the country with the highest mortality rate despite the early adoption of strict lockdowns [2]. Until the end of COVID-19's global health emergency in May 2023, five waves of SARS-CoV-2 infections were observed in Peru. Among them, the third wave took place between January and March 2022 and was characterized by the predominance of Delta and Omicron variants [3]. As a result, SARS-CoV-2 infections spread rapidly, reaching the highest peak of reported cases nationwide to date, with over 60,000 daily cases [4]. Although reported deaths were much lower than in previous waves of infections, a considerable number of hospitalizations were recorded, reaching 1,000 hospitalizations per week [5].

Self-medication has been a global concern worldwide in recent decades and the COVID-19 pandemic increased the tendency to resort to this practice to prevent SARS-CoV-2 infection with the outcomes it entails [6,7]. The absence of definitive treatments or vaccines against SARS-CoV-2 during the first year of the pandemic led to the widespread dissemination of false information about medications via social media [8]. In Peru, the prevalence of self-medication to treat COVID-19 was reported to be between 33.9–51.3%, which is consistent with data from prior to the pandemic [7,9]. This behaviour was facilitated by the inadequate sale of prescription medications in pharmacies despite the regulations imposed by the Peruvian government [10]. Previous research has focused on evaluating the efficacy of different medications in the treatment of COVID-19 [11,12]. However, studies examining the impact of these agents on the risk of SARS-CoV-2 infection in healthy individuals are scarce, despite the widespread adoption of self-administered prophylactics and the biological plausibility of such an effect [13–15].

Understanding the association between self-medication and SARS-CoV-2 infection remains important, even though COVID-19 is no longer considered a public health emergency of international concern. Self-medication practices can still affect timely diagnosis, disease progression, and healthcare-seeking behaviour, which are critical for managing ongoing cases and preparing for future respiratory outbreaks [8]. Investigating this relationship provides valuable insights for public health strategies aimed at promoting rational medication use and improving early detection and management of infectious diseases.

Our hypothesis was that self-administered prophylactic agents could affect the likelihood of SARS-CoV-2 infection. Therefore, this study aimed to assess whether these practices were associated with SARS-CoV-2 infection among traditional market vendors from the Peruvian Central Highlands during the third wave of infections.

## Methods

### Study design

In this nested case-control study, we enrolled vendors from traditional markets registered in a health promotion program called "Mercado saludable" (Healthful market). This program was implemented by a multidisciplinary research team in collaboration with the health services from August 2021 to August 2022 in the two largest traditional markets ("Market A" and "Market B" in this paper) from the province of Huancayo in the Junin region located in the Peruvian Central Highlands (Fig 1). The program aimed to support vendors reduce their risk of SARS-CoV-2 infection and mortality by enhancing the management of comorbidities and promoting healthy nutrition practices.

The program comprised of three sequential phases conducted before, during, and after the third wave of SARS-CoV-2 infections. During the first phase, from August to December 2021, the program worked with market vendors to establish its objectives, and consultation rooms were set up in locations within each market as designated by the vendors. Also, health workers assigned to each market oversaw participant registration and monitoring throughout the program. Recruitment was conducted through direct communication from market authorities to all workers in each market. Participation was voluntary and based on convenience sampling, as individuals self-enrolled by presenting themselves at the consultation room from their respective markets. No formal rejections were recorded, although not all eligible workers chose to participate. Finally, registration involved an interviewer-administered questionnaire, a clinical examination, and a nasopharyngeal swab for an antigen SARS-CoV-2 test (Roche Diagnostics SL, Barcelona, Spain), along with voluntary validation of COVID-19 vaccination status using individual vaccination cards given by the Peruvian Ministry of Health. During the second phase, from January to March 2022, registered participants were approached and offered a second nasopharyngeal swab for Reverse Transcriptase Real-Time Polymerase Chain Reaction (RT-qPCR) analysis to diagnose SARS-CoV-2.

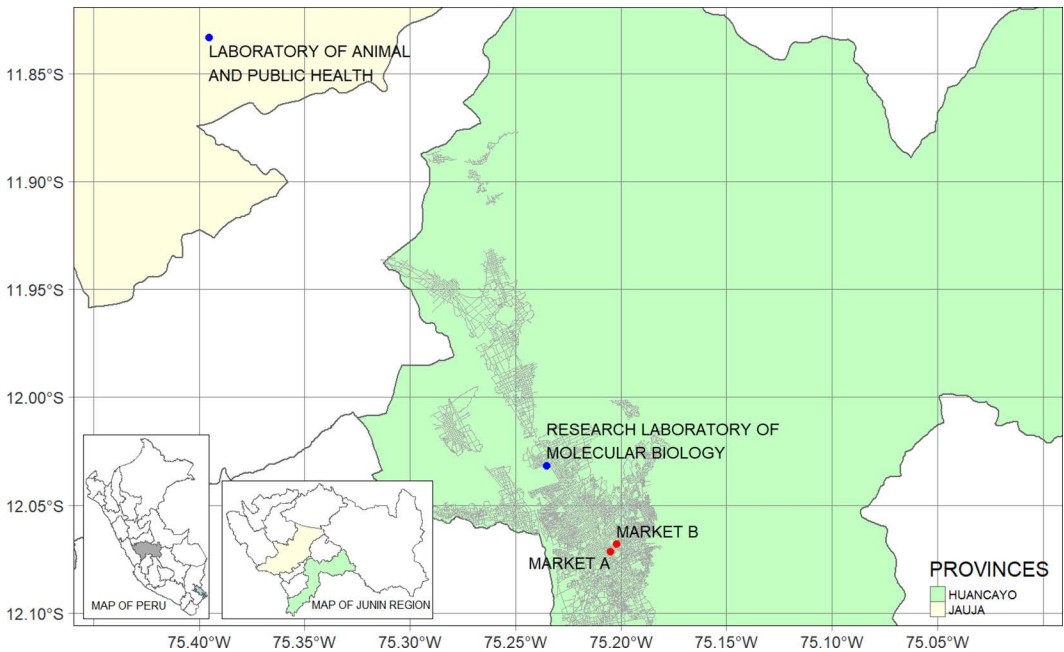

**Fig 1. Map of the province of Huancayo.** The main map presents the streets (dark grey lines) and points the studied wet markets (red dots) and laboratories (blue dots). The map of the Junin region presents the provinces of Huancayo (green) and Jauja (yellow), while the map of Peru presents the Junin region (dark grey). These maps were created using public domain basemaps provided by the "Instituto Nacional de Estadística e Informática" (https://ide.inei.gob.pe/) and the "Infraestructura de Datos Espaciales del Perú" (https://www.idep.gob.pe/).

Participants who provided consent were included in the following phase. During the third phase, from May to August 2022, participants underwent a third nasopharyngeal swab for SARS-CoV-2 diagnosis, and a blood sample was collected to evaluate haematological and biochemical parameters.

Participants with a positive SARS-CoV-2 diagnosis received their results individually and confidentially. These cases were immediately registered using the public health clinical inquiry form for COVID-19 provided by the Peruvian Ministry of Health. These sheets were filled out by the physician hired for this study and reported to the Junin Regional Health Directorate's Epidemiology Office. Moreover, these participants received free medical care and orientation from the health workers.

### Study population

The study cohort consisted of traditional market vendors who were enrolled in the program until its second phase ending in March 2022. The inclusion criteria were the following: 1) age 18 years or older, 2) a negative SARS-CoV-2 antigen test result at registration, and 3) known SARS-CoV-2 qPCR result at the second phase. Participants who did not validate their COVID-19 vaccination scheme with the vaccination card were excluded to minimize the chance of information bias.

### Cases and controls

Cases were defined as participants with a positive SARS-CoV-2 diagnosis through RT-qPCR during the third wave of infections, whereas the control group were participants with a negative test result. Participants with a positive SARS-CoV-2 diagnosis were randomly matched with control participants for age group, sex and market of origin. To minimize the reduction of statistical power during the analysis due to the exclusion of participants, cases were matched with controls using a variable case:control ratio up to a maximum of 1:3.

### SARS CoV-2 diagnosis through RT-qPCR

**Specimen collection.** For sample collection, sterile synthetic fiber swabs and vials with viral transport medium (HiMedia HiViral™ Transport Kit, Mumbai, India) were used. Before the procedure, the medical staff recorded the personal information (name, age, and sex) and relevant data (temperature, clinical signs, and number of vaccine doses applied) of each person. The patient was given a disposable towel and was told to expel as much nasal and oral secretion as possible to facilitate sample collection. The patient's head was placed at a 45° angle and a swab was inserted making circular movements through one nostril until it reached the posterior nasopharynx. The swab was kept inside for 10 seconds to obtain the most viral particles and was slowly withdrawn. Similarly, another swab was inserted into the mouth as far as the posterior pharynx, avoiding touching the tongue. Immediately after, both swabs were immersed in the transport medium. The samples were placed in an airtight container at 4°C and transported to the Laboratory of Animal and Public Health in the Tropical and Highlands Veterinary Research Institute of the National University of San Marcos located in El Mantaro district, region of Junin. The processing and analysis of the samples were carried out immediately after collection.

**Nucleic acid extraction and RT-qPCR.** Nucleic acid extraction was performed from 200 µL of each sample following the instructions of the manufacturer of the MagMAX™ Viral/Pathogen Nucleic Acid Isolation Kit (Applied Biosystems, Thermo Fisher Scientific, TX, USA). In addition, and to guarantee the correct development of the procedure, 200 µL of nuclease-free water were used as a negative control in each batch of samples processed. Finally, 50 µL of eluate was obtained from each sample and molecular analysis was performed immediately thereafter to avoid nucleic acid degradation associated with sample storage, freezing, and thawing.

The Centers for Disease Control and Prevention recommended RT-qPCR assay was used to simultaneously analyze and detect the N1 and N2 regions of the SARS-CoV-2 Nucleocapsid gene. The reaction mix was prepared in a final volume of 25 µL using 13.5 µL of SuperScript™ III Platinum™ One-Step qRT-PCR System (Invitrogen, Life Technologies, CA, USA), 0.5 µM of 2019-nCoV_N1 forward primer (5′-GACCCCAAAATCAGCGAAAT-3′), 0.5 µM 2019-nCoV_N1 reverse primer (5′-TCTGGTTACTGCCAGTTGAATCTG-3′), 0.2 µM 2019-nCoV_N1 probe (5′-/56-FAM/ACCCCGCAT/ ZEN/

TACGTTTGGTGGACC/3IABkFQ/-3´), 0.5 µM forward primer 2019-nCoV_N2 (5´-TTACAAACATTGGCCGCAAA-3´), 0.5 µM reverse primer 2019-nCoV_N2 (5´-GCGCGACATTCCGAAGAA-3´), 0.2 µM of 2019-nCoV_N2 probe (5´-/5SUN/ACAATTTGC/ZEN/CCCCAGCGCTTCAG/3IABkFQ/-3´), 3.5 µL of nuclease-free water, and 5 µL of RNA template [16]. All RT-qPCR assays were performed on a QuantStudio™ 3 real-time PCR machine (Thermo Fisher Scientific, USA) with the following temperature and time conditions: incubation at 25°C for 2 minutes followed by reverse transcription at 50°C for 15 minutes. Subsequently, initial denaturation at 95°C for 2 minutes and 45 cycles of denaturation at 95°C for 3 seconds followed by annealing and elongation at 55°C for 30 seconds [17]. A synthetic plasmid containing the N1 and N2 regions of the viral nucleocapsid (Integrated DNA Technologies, Inc., Coralville, IA, USA) was used as a positive control. Each sample was analyzed in duplicate and negative controls were included to validate the performance of the reaction. When nasopharyngeal samples exhibited cycle threshold values up to the 40th cycle during RT-qPCR assay, positive SARS-CoV-2 diagnosis was established. The extraction of nucleic acids and the preparation of the reaction mixtures were performed in different environments to avoid cross contamination.

### Questionnaire design

A structured interviewer-administered questionnaire was employed to collect data on participant's characteristics. The questionnaire was developed by the authors after a detailed review of the literature. As it was an ad hoc instrument, its content validity was evaluated by a panel of subject-matter experts appointed during the ethical review process. The experts assessed the questionnaire for clarity, relevance, and alignment with the study objectives. Minor revisions were made based on their feedback. Finally, a pilot test was conducted with vendors from an external traditional market to evaluate participants' comprehension and ensure the questions were appropriately understood. Additional minor adjustments were made based on insights gained from the pilot study to enhance clarity and consistency.

Apart from sociodemographic information such as age, sex and market of procedure, the questionnaire was composed of the following sections: 1) Medical record section, and 2) COVID-19 section (S1 Appendix). Participants were oriented by the health worker throughout the procedure. Finally, completed questionnaires were collected and entered in a Microsoft Excel sheet using ODK application [18]. Data was deidentified by codes to protect personal information.

### Clinical examination

The clinical examination was conducted after the completion of the questionnaire and consisted in a physical examination and a urine analysis. The physical examination evaluated the following parameters: 1) height (m), 2) weight (kg), 3) systolic blood pressure (mmHg), 4) diastolic blood pressure (mmHg), 5) temperature (°C), 6) oxygen saturation (%), 7) heart rate (beats per minute), 8) body fat percentage (%), 9) waist circumference (cm), and 10) glucose (mg/dL) using a glucometer. Body Mass Index (BMI) was calculated as weight in kilograms divided by height in meter squared. The results obtained in the clinical examination were collected in the same way as the information obtained in the questionnaire.

### Validation of COVID-19 vaccination scheme

During each phase of the study, participants voluntarily provided and validated information on the received vaccination scheme by accessing the COVID-19 vaccination card through the web portal developed by the Peruvian Ministry of Health [19]. Information on the number of doses received, fabricant of each vaccine, and date of vaccination were collected in a Microsoft Excel sheet.

### Self-administered prophylactic agents

Information on prophylactics adopted by participants to prevent COVID-19 was gathered from the questionnaire administered during the registration (S1 Appendix). Between August and December 2021, participants were asked to report whether or not they consumed the following agents to prevent COVID-19 during the pandemic as alternatives: 1)

acetylsalicylic acid, 2) ivermectin, 3) chlorine dioxide, 4) paracetamol, 5) ibuprofen, 6) corticosteroids, 7) azithromycin/clarithromycin, 8) penicillin/amoxicillin/ceftriaxone, 9) ciprofloxacin/levofloxacin, 10) enoxaparin, 11) N-acetylcysteine, 12) B complex, 13) vitamin C, 14) vitamin D, 15) zinc, and 16) others. As this question was presented as a multiple choice, dummy variables were created based on each one of the alternatives specified above. Finally, alternatives 7, 8 and 9 were merged into one variable denominated "antibiotics" due to the low numbers for specific antibiotic categories.

## Covariates

Two age groups were established: 18–59 years and 60 years and older, whereas information on comorbidities was collected through clinical examination conducted at registration. The diseases considered in this study include hypertension, obesity and diabetes due to their association with an increased risk of COVID-19. Hypertension was defined as having systolic blood pressure (SBP) higher than 139 or diastolic blood pressure (DBP) higher than 89. Obesity was defined as having a body mass index (BMI) of 30 or higher. Diabetes was defined as presenting a glucose level of 200 mg/dL or higher during the random blood sugar (RBS) test. Data on sex, market of origin and self-perceived risk of COVID-19 were directly obtained from the questionnaire (S1 Appendix).

Information on the vaccination program was collected from the COVID-19 vaccination card, including the number of doses administered, vaccine manufacturer, and vaccination date. For this study, information on vaccination was gathered until 14 days before the nasopharyngeal swab during the second phase to ensure effective protection against SARS-CoV-2 at the diagnosis. Given the diverse vaccination schemes in Peru, the immunization program was categorized based on whether participants had effective protection from COVID-19 booster which was available for the whole Peruvian population since November 2021. This categorization aligned with the booster dose recommendation to prevent SARS-CoV-2 infection by the Omicron variant, which was predominant during the third wave of infections in Peru.

## Data entry and statistical analysis

Participants were assigned unique random numbers to ensure anonymity during analysis while safeguarding personal information. Data from both the questionnaire and clinical examination conducted in the initial phase were collected on physical registration forms (S1 Appendix) and subsequently entered into a spreadsheet using ODK application. Conversely, information regarding SARS-CoV-2 diagnosis was directly entered into a separate spreadsheet. These databases were then integrated using the R program, version 4.2.2, merging them based on the unique identification code assigned to each participant. Afterward, the database underwent rigorous checks for completeness and was cleaned to ensure data quality.

To assess the association between categorical variables pertinent to the study objectives (self-administered prophylactics, COVID-19 booster, self-perceived risk of COVID-19, and comorbidities) and RT-qPCR-based SARS-CoV-2 diagnosis, the conditional logistic regression function from the R package "survival" was employed [20]. This allowed the estimation of both crude odds ratios (cORs) and adjusted odds ratios (aORs) with their corresponding 95% confidence intervals (95% CI). The adjusted model incorporated single predictors (self-administered prophylactics and COVID-19 booster) with a P value below 0.2 in the crude analysis, potential confounders (self-perceived risk of COVID-19 and comorbidities), and strong interaction effects between prophylactic practices and COVID-19 booster. Additionally, variables with fewer than 10 observations were excluded due to the substantial uncertainty reflected in coefficient estimates.

Regarding the adjusted model building, potential collinearity among the initial variables considered was evaluated using a correlation matrix based on Cramer's V coefficient, calculated with the "CramerV" function from the "DescTools" package in R [21]. If a group of variables presented at least a weak relationship (Cramer's V ≥ 0.2), the variable with weaker evidence of association with the SARS-CoV-2 diagnosis was discarded. Then, remaining variables were entered into a model without interactions following a direct approach. Subsequently, predictors with the weakest evidence of association were iteratively removed, assessing changes in estimated coefficients for remaining variables. This process was complemented by likelihood ratio tests, performed using the "lrtest" function from the "lmtest" package in R [22]. If estimated coefficients varied by 5% or more and the likelihood ratio test was

                                                                                      

significant (p-value<0.05), the predictor was deemed contributory to the final model and reintroduced. Otherwise, it was considered non-contributory and discarded. This process continued until only predictors with strong evidence of association and significant likelihood ratio test remained. Subsequently, interaction effects between each drug and COVID-19 booster were introduced into the model. Interaction lacking evidence of association were sequentially removed based on the estimated P values and supported by non-significant likelihood ratio tests. The elimination of interaction effects stopped once only those with strong evidence of association remained in the final adjusted model. Furthermore, the Akaike Information Criterion (AIC) was calculated for each adjusted model assessed using the "AIC" function from the "stats" package in R [23]. For this study, a P value less than 0.05 was considered to be strong evidence of association, whereas a P value less than 0.1 was interpreted as weak evidence of association.

### Ethics

This study was reviewed and approved by the Institutional Ethics Committee in Research (Comité Institucional de Ética en Investigación (CIEI)) at the Universidad Peruana Cayetano Heredia (certificate no. 676-24-21). Participants gave their informed consent before joining the study and their privacy was not compromised.

## Results

### Enrolment into the health promotion program and incidence of SARS-CoV-2 infection

Four hundred and thirty-nine participants were enrolled into the health promotion program. Upon enrolment, between August and December 2021, they underwent a SARS-CoV-2 antigen test, which was negative for 99.3% (436/439) of participants. In the second phase, conducted during the third wave of infections in Peru between January and March 2022, 254 out of the 439 initially registered participants (57.9%) consented to a nasopharyngeal swab for SARS-CoV-2 diagnosis through RT-qPCR analysis. Notably, the three participants who tested positive for SARS-CoV-2 in the first phase were not included among those 254 participants. As a result, 28.7% (73/254) were found to be positive. Finally, at the third phase, 186 participants with a RT-qPCR-based test in the second phase agreed to be tested for SARS-CoV-2 through the nasopharyngeal swab (73.2% of those at the second phase), with all participants testing negative. A flowchart showing the number of participants involved at each stage of the program is presented in Fig 2.

### Participants included in the analysis

For this study, all 73 participants who were found to be positive by SARS-CoV-2 RT-qPCR during the third wave of infections were included as cases. Controls were selected from the 181 participants with a negative test result at the same sampling round and randomly paired to the cases by sex, age group and market of origin. As a result, 176 controls were randomly matched presenting the following distribution: 32 cases had three controls, 39 cases had two controls, and two cases had one control. Among the 249 participants included, 70.7% (176/249) were female, 83.1% (207/249) were under 60 years old, and 50.6% (126/249) were from Market B.

### COVID-19 booster against SARS-CoV-2 infection

Based on the information collected from the vaccination cards, a broad range of applied immunization programs were found among participants (S1 Table). As shown in Table 1, the crude analysis shows that having an effective protection from the booster dose was associated with higher odds of SARS-CoV-2 infection during the third wave (cOR 3.02, 95% CI 1.41–6.50).

### Self-administered prophylactic agents for SARS-CoV-2 infection

As shown in Table 2, 87,2% (217/249) of the participants reported a wide variety of agents used to prevent SARS-CoV-2 infection. The most reported were paracetamol with 79.7% (173/217), ivermectin with 72.4% (157/217), ibuprofen with 58.1% (126/217), and vitamin C with 51.2% (111/217). Besides, 91.7% (199/217) of those who used self-administered

prophylactics reported combining two or more agents to prevent SARS-CoV-2 infection. Given that the self-administered agents scenario in this study is quite complicated and that individuals reported using a wide range of different combinations, a table describing each combination can be encountered in S2 Table.

In the crude analysis, only vitamin C consumption to prevent SARS-CoV-2 was associated with 49% reduced odds of getting a SARS-CoV-2 infection (cOR 0.51, 95% CI 0.28–0.93). Conversely, N-acetylcysteine consumption presented weak evidence of association with higher odds of getting a SARS-CoV-2 infection (cOR 3.06, 95% CI 0.86–10.98). Finally, acetylsalicylic acid consumption presented a P value less than 0.2 (see Table 1). For these reasons, consumption of these three agents were considered for the adjusted analysis.

## Self-perceived risk of COVID-19

At the time of registration for the health promotion program, 47.8% of participants (119/249) reported a medium level of self-perceived risk of contracting COVID-19. In comparison, 25.7% (64/249) reported a low level of perceived risk, 18.9% (47/249) reported a high level, and 7.6% (19/249) reported no perceived risk of contracting the disease. During the crude analysis, no association was found between levels of self-perceived risk and the odds of SARS-CoV-2 infection (Table 1).

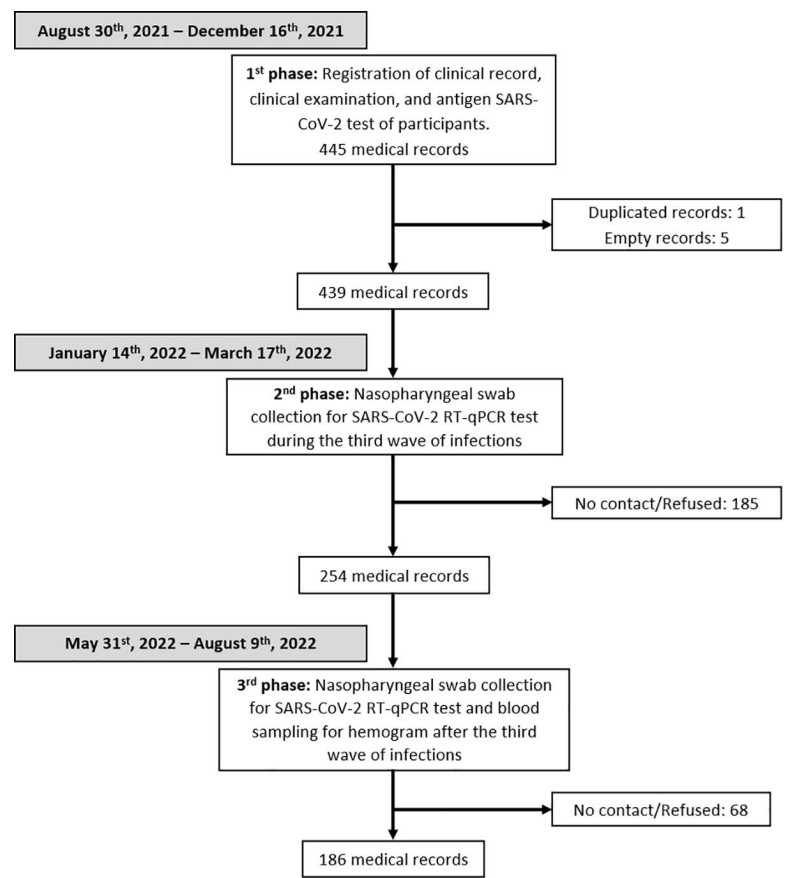

**Fig 2. Flowchart of the number of participants at each phase of the program.**

**Table 1. Crude and adjusted odds ratios (95% confidence interval) for SARS-CoV-2 infection through conditional logistic regression analysis.**

| Variables | COVID-19 diagnosis through RT-qPCR | | Conditional logistic regression | | | |
| --- | --- | --- | --- | --- | --- | --- |
| | | | Crude analysis[a] | | Adjusted analysis[b] | |
| | Positive n = 73 (%) | Negative n = 176 (%) | cOR (95% CI) | p-value | aOR (95% CI) | p-value |
| **Demographics** | | | | | | |
| **Sex** | | | | | | |
| Female | 49 (67.1%) | 127 (72.2%) | – | | – | |
| Male | 24 (32.9%) | 49 (27.8%) | – | – | – | – |
| **Age group** | | | | | | |
| 18–59 years | 61 (83.6%) | 146 (83.0%) | – | | – | |
| 60 years or more | 12 (16.4%) | 30 (17.0%) | – | – | – | – |
| **Market of procedure** | | | | | | |
| Market A | 34 (46.6%) | 89 (50.6%) | – | | – | |
| Market B | 39 (53.4%) | 87 (49.4%) | – | – | – | – |
| **Comorbidities** | | | | | | |
| **Hypertension** | | | | | | |
| No (SBP < 140 and DBP < 90) | 70 (95.9%) | 151 (85.8%) | Ref. | | Ref. | |
| Yes (SBP ≥ 140 or DBP ≥ 90) | 3 (4.1%) | 25 (14.2%) | 0.27 (0.08–0.92) | 0.037* | 0.21 (0.06–0.78) | 0.020* |
| **Obesity** | | | | | | |
| No (BMI < 30) | 56 (76.7%) | 129 (73.3%) | Ref. | | Ref. | |
| Yes (BMI ≥ 30) | 17 (23.3%) | 47 (26.7%) | 0.94 (0.49–1.79) | 0.848 | 1.01 (0.47–2.17) | 0.986 |
| **Diabetes** | | | | | | |
| No (RBS < 200 mg/dl) | 73 (100.0%) | 174 (98.9%) | NC | | – | |
| Yes (RBS ≥ 200 mg/dl) | 0 (0.0%) | 2 (1.1%) | | | – | – |
| **Self-perceived risk of COVID-19** | | | | | | |
| High | 10 (13.7%) | 37 (21.0%) | Ref. | | Ref. | |
| Medium | 34 (46.6%) | 85 (48.3%) | 1.62 (0.66–4.00) | 0.296 | 2.08 (0.76–5.70) | 0.153 |
| Low | 23 (31.5%) | 41 (23.3%) | 2.27 (0.85–6.09) | 0.104 | 2.76 (0.92–8.30) | 0.070 |
| None | 6 (8.2%) | 13 (7.4%) | 1.57 (0.42–5.84) | 0.504 | 1.56 (0.38–6.41) | 0.539 |
| **COVID-19 booster** | | | | | | |
| Yes (3 doses) | 11 (15.1%) | 58 (33.0%) | Ref. | | Ref. | |
| No (0–2 doses) | 62 (84.9%) | 118 (67.0%) | 3.02 (1.41–6.50) | 0.005* | 3.38 (1.43–7.95) | 0.005* |
| **Self-medication practices to prevent SARS-CoV-2 infection** | | | | | | |
| **None** | | | | | | |
| No (≥ 1 medication) | 66 (90.4%) | 151 (85.8%) | Ref. | | – | |
| Yes (0 medications) | 7 (9.6%) | 25 (14.2%) | 0.67 (0.28–1.64) | 0.382 | – | – |
| **Acetylsalicylic acid** | | | | | | |
| No | 50 (68.5%) | 138 (78.4%) | Ref. | | Ref. | |
| Yes | 23 (31.5%) | 38 (21.6%) | 1.53 (0.86–2.73) | 0.146 | 2.34 (1.17–4.66) | 0.016* |
| **Ivermectin** | | | | | | |
| No | 25 (34.2%) | 67 (38.1%) | Ref. | | – | |
| Yes | 48 (65.8%) | 109 (61.9%) | 1.23 (0.66–2.23) | 0.500 | – | – |
| **Chlorine dioxide** | | | | | | |
| No | 67 (91.8%) | 160 (90.9%) | Ref. | | – | |
| Yes | 6 (8.2%) | 16 (9.1%) | 0.78 (0.29–2.11) | 0.626 | – | – |
| **Paracetamol** | | | | | | |
| No | 18 (24.7%) | 58 (33.0%) | Ref. | | – | |
| Yes | 55 (75.3%) | 118 (67.0%) | 1.48 (0.81–2.69) | 0.201 | – | – |

*(Continued)*

**Table 1.** (Continued)

| Variables | COVID-19 diagnosis through RT-qPCR | | Conditional logistic regression | | | |
|---|---|---|---|---|---|---|
| | | | Crude analysis[a] | | Adjusted analysis[b] | |
| | Positive n = 73 (%) | Negative n = 176 (%) | cOR (95% CI) | p-value | aOR (95% CI) | p-value |
| **Ibuprofen** | | | | | | |
| No | 34 (46.6%) | 89 (50.6%) | Ref. | | – | |
| Yes | 39 (53.4%) | 87 (49.4%) | 1.14 (0.67–1.95) | 0.633 | – | – |
| **Corticosteroids** | | | | | | |
| No | 57 (78.1%) | 137 (77.8%) | Ref. | | – | |
| Yes | 16 (21.9%) | 39 (22.2%) | 0.97 (0.50–1.89) | 0.933 | – | – |
| **Antibiotics** | | | | | | |
| No | 40 (54.8%) | 105 (59.7%) | Ref. | | – | |
| Yes | 33 (45.2%) | 71 (40.3%) | 1.22 (0.71–2.10) | 0.472 | – | – |
| **Enoxaparin** | | | | | | |
| No | 72 (98.6%) | 174 (98.9%) | Ref. | | – | |
| Yes | 1 (1.4%) | 2 (1.1%) | 1.30 (0.12–14.51) | 0.830 | – | – |
| **N-acetylcysteine** | | | | | | |
| No | 67 (91.8%) | 171 (97.2%) | Ref. | | – | |
| Yes | 6 (8.2%) | 5 (2.8%) | 3.06 (0.86–10.98) | 0.086 | – | – |
| **B complex** | | | | | | |
| No | 54 (74.0%) | 119 (67.6%) | Ref. | | – | |
| Yes | 19 (26.0%) | 57 (32.4%) | 0.68 (0.36–1.28) | 0.232 | – | – |
| **Vitamin C** | | | | | | |
| No | 48 (65.8%) | 90 (51.1%) | Ref. | | Ref. | |
| Yes | 25 (34.2%) | 86 (48.9%) | 0.51 (0.28–0.93) | 0.027* | 0.44 (0.23–0.87) | 0.017* |
| **Vitamin D** | | | | | | |
| No | 57 (78.1%) | 135 (76.7%) | Ref. | | – | |
| Yes | 16 (21.9%) | 41 (23.3%) | 0.80 (0.39–1.64) | 0.544 | – | – |
| **Zinc** | | | | | | |
| No | 61 (83.6%) | 145 (82.4%) | Ref. | | – | |
| Yes | 12 (16.4%) | 31 (17.6%) | 0.89 (0.42–1.88) | 0.753 | – | – |
| **Others** | | | | | | |
| No | 67 (91.8%) | 159 (90.3%) | Ref. | | – | |
| Yes | 6 (8.2%) | 17 (9.7%) | 0.88 (0.33–2.37) | 0.805 | – | – |

SBP: Systolic blood pressure, DBP: diastolic blood pressure, BMI: Body mass index, RBS: Random blood sugar, NC: Not computable

[a]Simple regression model adjusted by variables Sex, Age group, and Market of procedure.

[b]Multiple regression model: Hypertension + Obesity + Self-perceived COVID-19 risk + Vitamin C + Acetylsalicylic acid + COVID-19 booster.

*P-value less than 0.05.

## Comorbidities in participants

Regarding comorbidities detected through clinical examination, 11.3% (28/249) of participants had blood pressure levels indicative of hypertension. In crude analysis, hypertension was associated with a 73% reduction in the odds of contracting a SARS-CoV-2 infection (crude odds ratio [cOR] 0.27, 95% confidence interval [CI] 0.08–0.92). Additionally, 25.7% (64/249) of participants were diagnosed with obesity. Finally, only 0.8% (2/249) had random blood sugar (RBS) test results consistent with uncontrolled diabetes.

**Table 2. Medication consumption to prevent SARS-CoV-2 infection among participants.**

| Variables | Number of participants | Percentage of total number of participants (n = 249) |
|---|---|---|
| **Medications** | | |
| None | 32 | 12.9% |
| Acetylsalicylic acid | 61 | 24.5% |
| Ivermectin | 157 | 63.1% |
| Chlorine dioxide | 22 | 8.8% |
| Paracetamol | 173 | 69.5% |
| Ibuprofen | 126 | 50.6% |
| Corticosteroids | 55 | 22.1% |
| Antibiotics | 104 | 41.8% |
| Enoxaparin | 3 | 1.2% |
| N-acetylcysteine | 11 | 4.4% |
| B complex | 76 | 30.5% |
| Vitamin C | 111 | 44.6% |
| Vitamin D | 57 | 22.9% |
| Zinc | 43 | 17.3% |
| Others | 23 | 9.2% |
| **Consumption of groups of medications** | | |
| One medication | 18 | 7.2% |
| Two medications | 29 | 11.6% |
| Three medications | 33 | 13.3% |
| Four medications | 29 | 11.6% |
| Five medications | 33 | 13.3% |
| Six medications | 25 | 10.0% |
| Seven medications | 17 | 6.8% |
| Eight medications | 14 | 5.6% |
| Nine medications | 8 | 3.2% |
| Ten medications | 8 | 3.2% |
| Eleven medications | 3 | 1.2% |

## Adjusted analysis

For the adjusted analysis, COVID-19 booster, and the use of prophylactic agents (vitamin C, N-acetylcysteine, and acetyl-salicylic acid) were initially considered as predictors. Also, self-perceived risk of COVID-19 and comorbidities were considered as confounding variables. However, diabetes detected through clinical examination was removed due to presenting fewer than 10 observations. N-acetylcysteine consumption was also excluded based on the detected collinearity with vitamin C through correlation matrix analysis (S3 Table). The remaining variables did not exhibit potential collinearity, as all Cramér's V coefficients were below 0.2. Thus, the initial model without interaction effects included COVID-19 booster, vitamin C consumption, acetylsalicylic acid consumption, self-perceived risk of COVID-19, hypertension, and obesity. This allowed estimation of main effects with wide variation in their presence among participants (Fig 3).

When evaluating the complete model without interaction effects, it was found that all predictors presented strong evidence of association, and these provided significant information to the model through likelihood ratio test (S2 Appendix). Also, comorbidities minimally impacted the estimated coefficients (never exceeding 5%) but were retained as potential confounders. Therefore, it was decided not to remove any variable from the model and to continue evaluating the interaction effects between each medication consumption (vitamin C and acetylsalicylic acid) and COVID-19 booster. As a result,

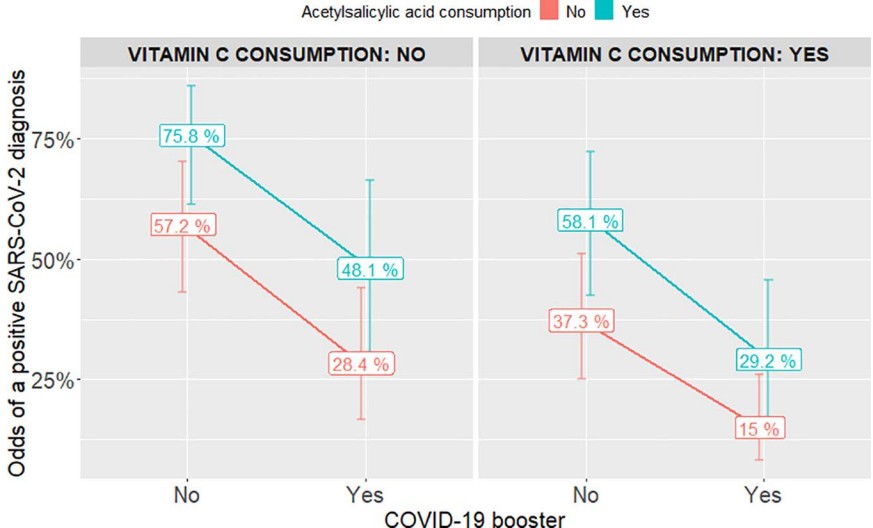

**Fig 3. Distribution of participants by variables presenting the main effects of the analysis.**

it was found that no interaction effect presented strong evidence of association and did not provide significant information to the model through likelihood ratio test (S2 Appendix). Thus, it was decided to maintain the model without interaction effects as the final adjusted model which presented an AIC of 164.6. Therefore, not having the COVID-19 booster was associated with 3.38-fold increased odds of SARS-CoV-2 infection (aOR 3.38, 95% CI 1.43–7.95). Regarding self-administered prophylactics, acetylsalicylic acid consumption was associated with 2.34-fold increased odds of SARS-CoV-2 infection in participants (aOR 2.34, 95% CI 1.17–4.66). Finally, vitamin C consumption maintained a protective effect, being associated with 56% reduced odds of SARS-CoV-2 infection (aOR 0.44, 95% CI 0.23–0.87). Multivariable analysis results are shown in Fig 4.

**Fig 4. Predicted probabilities calculated through the adjusted conditional logistic regression model.** The x axis represents the COVID-19 booster dose and y axis the predicted odds of a positive SARS-CoV-2 diagnosis. Lines represent participants with (blue) or without (red) acetylsalicylic acid consumption. Columns represent the categorization by vitamin C consumption. The lowest risk of positive test result is observed in the group with booster dose, no consumption of acetylsalicylic acid and consumption of vitamin C.

## Discussion

To our knowledge, this nested case-control study is the first to explore the hypothesis that self-administered prophylaxis, within a real-world context, are associated with the odds of contracting SARS-CoV-2. Our study revealed that vendors in traditional markets employed a wide range of agents to prevent SARS-CoV-2 infection. Even in fully vaccinated participants, the only agent that was highly associated to infection was acetylsalicylic acid intake. These elevated odds matched those obtained for individuals without the COVID-19 booster. Additionally, the results of the analysis provide evidence of an association between consumption of vitamin C and lower odds of contracting SARS-CoV-2 infection. Thus, the impact of self-administered prophylaxis during the wave with the highest peak of reported cases in Peru is supported by "real world evidence" according to our findings. On the other side, our study corroborates the association between COVID-19 booster and reduced odds of SARS-CoV-2 infection, consistent with the documented efficacy of boosters against the Delta and Omicron variants [24]. These variants predominated during the third wave of COVID-19 infections in Peru [3].

The data used in our analysis were collected between August 2021 and March 2022, a period during which Peru experienced distinct phases of the COVID-19 pandemic, marked by fluctuations in transmission rates, vaccination coverage, and public health responses. At the onset of the study, in the latter half of 2021, Peru was emerging from its second wave of COVID-19, which had been associated with the highest mortality rate and was primarily driven by the Lambda and Gamma variants [3]. By this time, daily reported cases had declined to an average of approximately 1,000 per day [4]. Concurrently, vaccine availability was beginning to expand, and several public health restrictions remained in effect [25]. These factors may have contributed to the elevated perception of COVID-19 risk observed among participants (66.7% reported a medium to high level of perceived risk). This context may also help explain the high presence of self-medication practices for COVID-19 prevention found in our study, which exceeded previously reported self-medication prevalence up to 51.3% in Peru [7,9].

In contrast, a markedly different context was observed during the second phase of our study in early 2022. At this point, Peru was entering its third wave of COVID-19, characterized by the highest peak of daily reported cases (nearly 60,000), although accompanied by considerably lower mortality rates. This wave was predominantly driven by the Delta and Omicron variants [3]. Daily cases numbers rose sharply from the previously reported average of 1,000 [4]. Additionally, COVID-19 booster doses had become widely available for adults across all age groups, and many restrictions had been relaxed, allowing for a gradual return to pre-pandemic routines [26]. Despite the availability of booster doses, fewer than one-third of participants were protected with the COVID-19 booster dose at the time of their diagnosis during the second phase of our study. This low coverage may be partially explained by reduced vaccine acceptance in several regions of the Central Highlands in Peru [27]. Bendezu-Quispe *et al.* [28] suggest that lower educational attainment may contribute to this pattern, as their study found that not having a postgraduate degree was associated with lower uptake of the third dose. This is particularly relevant given that education indicators in the Highlands regions tend to be lower than those in the coastal areas [29]. Furthermore, the phenomenon known as "pandemic fatigue" -characterized by diminished motivation to adhere to health measures over time- may have contributed to reduced adherence to full vaccination schedules [30,31]. Although the efficacy of the third dose has been demonstrated in various studies [32–35], including our own findings, addressing the underlying causes of reduced vaccination intent remains essential to improving the effectiveness of future immunization campaigns.

While acetylsalicylic acid has shown a reduced in-hospital mortality among COVID-19 patients [12,36], evidence regarding its efficacy in preventing SARS-CoV-2 infection remains limited and inconsistent. A retrospective population-based cross-sectional study conducted in Israel suggested a potential link between acetylsalicylic acid consumption for cardiovascular disease prevention and a reduced likelihood of SARS-CoV-2 infection [37]. However, a systematic review found no such association between prior non-steroidal anti-inflammatory drug use, including acetylsalicylic acid, and SARS-CoV-2 infection odds [38]. It should be considered that these studies were conducted primarily among elderly patients during the early stages of the COVID-19 pandemic, characterized by strict movement restrictions aimed at

reducing disease spread. Our study, however, reveals a contrasting finding: an association between acetylsalicylic acid consumption for SARS-CoV-2 prevention and increased infection odds. This finding may be related to the proposed mechanism of upregulation of the angiotensin-converting enzyme 2 (ACE-2), host receptor for SARS-CoV-2, which could increase SARS-CoV-2 infectivity [15]. Although this upregulation process has not been proven in healthy subjects, this observational study might be the first insight of this. On the other hand, this finding could be attributed to the fact that participants who take acetylsalicylic acid perceive themselves to be at a higher risk of severe COVID-19 because they also perceive themselves to be at risk for cardiovascular disease [39,40]. In this manner, they may believe that reducing their perceived risk of cardiovascular disease through acetylsalicylic acid could protect them from COVID-19, leading to low adherence to safety practices such as social distancing [41]. Nonetheless, a comprehensive understanding of how acetylsalicylic acid contributes to SARS-CoV-2 infectivity in healthy individuals necessitates further investigation in future studies.

Besides, our study also found association between vitamin C consumption and reduced odds of SARS-CoV-2 infection. This finding agrees with the proposed potential of this supplement to prevent COVID-19 infection [42]. Apart from boosting immune response, vitamin C has been shown to facilitate the degradation of ACE-2, providing protection against SARS-CoV-2 infection using an animal model [14]. Although the inclusion of vitamin C in the treatment of COVID-19 does not seem to have a beneficial effect according to recent clinical trials [11,43], the administration and anticipated outcomes of this supplement vary between hospitalized and healthy individuals. Although it was demonstrated that vitamin C supplementation could reduce only in 4% the risk of acute respiratory infection [44], no studies on the prevention of COVID-19 using vitamin C in healthy subjects were found during the literature review as previously reported [45]. As far as we know, this is the first study which describes a protective effect of vitamin C consumption against SARS-CoV-2 infection among healthy subjects in a real-world setting. Nevertheless, clinical trials assessing this topic are needed to establish a more robust association since SARS-CoV-2 is constantly mutating and novel variants will appear continuously leading to future COVID-19 outbreaks [46].

Blood pressure measurement compatible with hypertension was associated with lower odds of SARS-CoV-2 infection. Although there is no evidence of direct association between hypertension and a high likelihood of SARS-CoV-2 infection, hypertension is the most common comorbidity reported in COVID-19 patients and a significant risk factor for COVID-19 hospitalization and mortality [47]. Therefore, it is possible that participants with hypertension maintained stricter safety practices to prevent COVID-19.

Our study has limitations. First, the results found in our study may not be fully generalizable, as participants were exclusively traditional market vendors, and recruitment was conducted through a non-random, convenience-based approach. Specifically, participation was voluntary and limited to individuals who actively approached the registration office in response to announcements made by market authorities. This self-selection process may have led to selection bias, as those more interested or concerned about COVID-19 might have been more likely to enroll. In addition, no formal records of refusals during the first phase of the health program, further limiting the ability to assess the direction or magnitude of this potential bias. Nonetheless, according to the last National Census of Food Markets in 2016 [48], Market A reported approximately 600 operational stalls and Market B reported 1,000. In comparison, our study initially recruited 224 vendors from Market A and 215 from Market B, suggesting that a substantial fraction of the total vendor population was recruited although the obtained sample could be potentially biased. Further, traditional markets could still serve as potential SARS-CoV-2 sentinel surveillance sites, given their constant interaction with the general population.

Second, our data collection occurred during two markedly different scenarios of the COVID-19 pandemic, as previously described. The information on perceived risk of COVID-19 and self-medication practices was gathered during a period characterized by a high uncertainty regarding vaccine availability and following the most lethal wave of infections and responses were not reconfirmed at later stages of the program. By contrast, participants were diagnosed with SARS-CoV-2 during the third wave, a time when vaccines were widely available and mortality rates had considerably declined.

Consequently, it is possible that participants perceptions and behaviours had changed by the time of their diagnosis. This temporal discrepancy may have led to an overestimation of perceived risk and self-medication practices, which should be considered when interpreting the findings of this study.

Third, SARS-CoV-2 diagnosis was performed using two different methods due to logistical constraints. According to a systematic review on rapid antigen tests for SARS-CoV-2 [49], the test used in our study reported a sensitivity and specificity of 92.5% and 97.8% when compared with PCR assay, respectively [50]. This difference in diagnostic accuracy may have led to misclassification, particularly underestimating infections during the pre-third wave phase due to the lower sensitivity of the antigen test compared to RT-qPCR. As a result, some participants classified as SARS-CoV-2 negative before the third wave could have been false negatives, which might have affected the estimation of associations between self-administered prophylaxis and subsequent infection. This potential bias should be considered when interpreting the study findings.

Fourth, self-administered prophylaxis data relied on self-report, introducing recall bias. To mitigate this, interviewers were trained to guide participants through questionnaire completion. Fifth, the self-administered prophylaxis questions assessed only the occurrence of consumption, without capturing details on dosage and frequency. This limitation may introduce information bias, as the dosage and frequency of medication use likely varied widely among participants. Consequently, it is not possible to establish a robust causal relationship between reported consumption and SARS-CoV-2 diagnosis. This should be considered when interpreting the results and highlights the need for more detailed assessments in future research.

Sixth, the questionnaire used was an ad hoc instrument developed specifically for the purposes of this study. Although it underwent expert review as part of the ethical assessment process, and was piloted in a group of vendors from an external traditional market to evaluate clarity and participant understanding, the questionnaire may still present limitations in its psychometric robustness and could introduce measurement bias in self-reported behaviours. Lastly, certain self-administrated agents may be controversial, potentially leading to conformity bias. Nevertheless, interviewers assured participants of response anonymity and non-impact on program continuity.

In conclusion, despite these limitations, this nested case-control study shows that acetylsalicylic acid consumption to prevent SARS-CoV-2 increased the infection odds during the third wave in Peru. Conversely, vitamin C consumption with the same purpose had a protective effect against the SARS-CoV-2 infection. The results obtained in real-world settings from our study indicate that further studies on the use of these medications to prevent SARS-CoV-2 infection enrolling healthy subjects are needed to establish a strong causal relationship.

## Supporting information

**S1 Appendix. Questionnaire administered at the registration of the "Mercado saludable" program.**
(PDF)

**S2 Appendix. Building process of conditional logistic regression adjusted model.**
(PDF)

**S1 Table. SARS-CoV-2 diagnosis by vaccination scheme in selected participants.**
(PDF)

**S2 Table. Combination of medications consumed to prevent SARS-CoV-2 infection reported by participants.**
(PDF)

**S3 Table. Correlation matrix analysis with Cramer's V coefficient for the adjusted conditional logistic regression model building.**
(PDF)

**S4 Table. Data from the "Mercado saludable" program analyzed in the study.**
(CSV)

## Acknowledgments

We thank Dr. Ruby Chang from the Department of Comparative Biomedical Sciences of the Royal Veterinary College for her support in reviewing the statistical analysis applied in this study.

## Author contributions

**Conceptualization:** Ana Ho-Palma, Christine Leyns, Javier Guitian, Eloy Gonzales-Gustavson.

**Data curation:** Daniel A. Andrade, Ana Ho-Palma, Cesar A. Valdivia-Carrera.

**Formal analysis:** Daniel A. Andrade, Ana Ho-Palma, Cesar A. Valdivia-Carrera.

**Funding acquisition:** Javier Guitian, Eloy Gonzales-Gustavson.

**Investigation:** Daniel A. Andrade, Ana Ho-Palma, Cesar A. Valdivia-Carrera.

**Methodology:** Daniel A. Andrade, Ana Ho-Palma, Cesar A. Valdivia-Carrera, Astrid Munguia, Christine Leyns, Javier Guitian, Eloy Gonzales-Gustavson.

**Project administration:** Javier Guitian, Eloy Gonzales-Gustavson.

**Resources:** Javier Guitian, Eloy Gonzales-Gustavson.

**Supervision:** Ana Ho-Palma, Christine Leyns, Javier Guitian, Eloy Gonzales-Gustavson.

**Validation:** Daniel A. Andrade, Ana Ho-Palma, Cesar A. Valdivia-Carrera, Astrid Munguia, Christine Leyns, Javier Guitian, Eloy Gonzales-Gustavson.

**Visualization:** Daniel A. Andrade, Astrid Munguia, Christine Leyns.

**Writing – original draft:** Daniel A. Andrade.

**Writing – review & editing:** Daniel A. Andrade, Ana Ho-Palma, Cesar A. Valdivia-Carrera, Astrid Munguia, Christine Leyns, Javier Guitian, Eloy Gonzales-Gustavson.

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
