## [Decision Letter · Decision Letter 0]

Association between self-administrated prophylactics and SARS-CoV-2 infection among traditional market vendors from the Central Highlands of Peru: A nested case-control study

PLOS ONE

Dear Dr. Guitian,

Thank you for submitting your manuscript to PLOS ONE. After careful consideration, we feel that it has merit but does not fully meet PLOS ONE’s publication criteria as it currently stands. Therefore, we invite you to submit a revised version of the manuscript that addresses the points raised during the review process.

The manuscript presents a valuable contribution to understanding prophylactic medication use during the COVID-19 pandemic through a well-designed case-control study. After careful review of the manuscript and reviewers' comments, three key modifications are required for acceptance:

The statistical validation must be strengthened by including appropriate goodness-of-fit measures for the conditional logistic regression and assessing collinearity among predictor variables, particularly given the unexpected 2.13-fold increased odds with acetylsalicylic acid use. The authors must address potential indication bias by incorporating the collected perceived risk data (S1 Appendix, Q9) into the analysis, or if not feasible, provide explicit justification and discuss this limitation. Finally, a clear temporal context describing the relevant COVID-19 wave in Peru during the study period is needed for proper interpretation, particularly for international readers.

My additional recommendations and observations can be found below.

We look forward to receiving your revised manuscript.

Kind regards,

Yordanis Enríquez Canto, Ph.D.

Academic Editor

PLOS ONE

“This work was supported by the Medical Research Council and the National Institute for Health and Care Research (NIHR) [MR/V028561/1]. DA and EG received partial support of the Fogarty International Center/NIH [Training grant number D43TW00114]. DA also received partial support of the US Centers for Disease Control and Prevention cooperative agreement [Award number 5U01GH00266].”

3. We note that Figure 3 in your submission contain copyrighted images. All PLOS content is published under the Creative Commons Attribution License (CC BY 4.0), which means that the manuscript, images, and Supporting Information files will be freely available online, and any third party is permitted to access, download, copy, distribute, and use these materials in any way, even commercially, with proper attribution. For more information, see our copyright guidelines: http://journals.plos.org/plosone/s/licenses-and-copyright.

a. You may seek permission from the original copyright holder of Figure 3 to publish the content specifically under the CC BY 4.0 license.

5. We note that there is identifying data in the Supporting Information file <S4_table.csv>. Due to the inclusion of these potentially identifying data, we have removed this file from your file inventory. Prior to sharing human research participant data, authors should consult with an ethics committee to ensure data are shared in accordance with participant consent and all applicable local laws.

-Location data

Please remove or anonymize all personal information, ensure that the data shared are in accordance with participant consent, and re-upload a fully anonymized data set. Please note that spreadsheet columns with personal information must be removed and not hidden as all hidden columns will appear in the published file.

Additional Editor Comments:

To further strengthen this valuable contribution, we recommend addressing several key points:

First, regarding essential statistical validation, as noted by Reviewer 2, additional model diagnostics would enhance the findings. We recommend including appropriate goodness-of-fit measures for the conditional logistic regression and assessing potential collinearity among key predictor variables, particularly given the significant associations reported with prophylactic medications.

Second, with respect to risk perception and indication bias, the perceived COVID-19 risk data collected (S1 Appendix, Q9) could help address potential indication bias related to prophylactic medication use. If feasible, we suggest incorporating the perceived risk variable into the analysis to address this bias, as individuals at higher risk may be more likely to take preventive medications. This incorporation could also help explain the unexpected association with acetylsalicylic acid use and understand how risk perception influences both medication use and preventive behaviors. At a minimum, please describe the distribution of risk perception between cases and controls.

Third, as Reviewer 1 noted, enhancing the context would benefit international readers. We recommend adding a brief section that describes the relevant characteristics of COVID-19 waves in Peru during the study period and explains how this context might influence result interpretation. This addition would significantly improve the paper's utility for international audiences.

Next, in the limitations discussion, your current section appropriately addresses the binary nature of prophylaxis data. We encourage you to maintain this discussion but consider briefly explaining how this limitation might affect the interpretation of the reported associations, along with suggesting key considerations for future research.

Lastly, we believe that clarifying the contemporary relevance of the findings could enhance the impact of your study. Strengthening the introduction with a brief overview of current public health implications and relevance to future pandemic preparedness would be beneficial.

Please prioritize the statistical enhancements suggested by Reviewer 2, while also considering the integration of perceived risk data based on availability and feasibility. Additionally, the temporal context addition need not be extensive, but it should provide sufficient information for proper interpretation.

Reviewers' comments:

Reviewer's Responses to Questions

**Comments to the Author**

1. Is the manuscript technically sound, and do the data support the conclusions?

Reviewer #1: Yes

Reviewer #2: Yes

2. Has the statistical analysis been performed appropriately and rigorously?

Reviewer #1: Yes

Reviewer #2: Yes

3. Have the authors made all data underlying the findings in their manuscript fully available?

Reviewer #1: Yes

Reviewer #2: Yes

4. Is the manuscript presented in an intelligible fashion and written in standard English?

Reviewer #1: Yes

Reviewer #2: Yes

Reviewer #1: We would like to commend the authors for their rigorous methodological approach to this study. While the study may have been significant during the COVID-19 pandemic, it is not entirely clear why its findings remain relevant in the present context. The introduction should explicitly justify why these data are still important today and how they contribute to current public health discussions.

Additionally, the study collected data between 2021 and 2022, yet a substantial portion of the survey responses reflect cross-sectional data from 2021. This is particularly important in the Peruvian context, as the characteristics of each COVID-19 wave differed significantly, especially in terms of population practices and behavioral trends. Although the authors acknowledge this as a limitation, further clarification on how this temporal variation may have influenced the results would be beneficial.

Furthermore, the study’s findings may not be fully generalizable, not only due to the specific population studied but also because different COVID-19 waves may have exhibited distinct epidemiological and behavioral patterns.

Another key limitation is that the questionnaire only assessed whether participants self-administered prophylactic treatments, without considering dosage or duration of use. This is a crucial factor, as some individuals may have taken prophylactics sporadically (once), while others may have engaged in prolonged or repeated use, leading to potentially different health implications. While it is understandable that the study focuses on self-administration, it is possible that these subgroups differed in ways that could affect the observed associations. This limitation should be further elaborated in the discussion, along with an explanation of how it could impact the study's results.

Lastly, we strongly recommend including a dedicated section that contextualizes the COVID-19 situation in Peru at the time of data collection. This would be particularly beneficial for international readers unfamiliar with the Peruvian pandemic timeline, as references to the "third wave" might otherwise be misinterpreted based on the experience of other countries, where variant predominance, timeframes, and vaccination coverage varied. Providing this background would enhance the clarity and applicability of the findings.

Finally, the study states that "the employed rapid antigen test has a sensitivity of 96.5% and a specificity of 99.7%." This assertion should be properly cited, as existing literature reports varying values for the sensitivity and specificity of rapid antigen tests. For example, a review published in PLOS presents different estimates. Proper citation of the source for these values is necessary to ensure scientific accuracy and allow readers to critically assess the diagnostic validity of the test used in the study.

Reviewer #2: El trabajo aplica correctamente la metodología, sin embargo, pudiera precisar cuál fue la bondad de ajuste para no sobreestimar los resultados como como el test de Hosmer-Lemeshow o AIC/BIC. No se reporta si hubo problemas con la colinealidad de las variables finales.

**Do you want your identity to be public for this peer review?** For information about this choice, including consent withdrawal, please see our Privacy Policy

Reviewer #1: No

Reviewer #2: No

---

## [Author Response · Author response to Decision Letter 1]

22 May 2025

Dear Editor,

We sincerely thank the reviewers and the editorial team for their thoughtful comments and suggestions, which have greatly contributed to improving our manuscript (PONE-D-25-09334). In response, we have revised the manuscript accordingly and are pleased to submit the updated version for your consideration.

All comments have been addressed in the accompanying document. Our responses are presented in green, and the corresponding edits to the manuscript are shown in red.

We would also like to offer our sincere apologies for a typographical error identified in Table 1. Specifically, the absolute frequencies for the “Hypertension” variable and the relative frequencies for the “Obesity” variable were incorrectly reported. This mistake was due to a mistyping and did not originate from the dataset itself. We have now corrected these values. Importantly, the rest of the analysis remains unaffected, as the underlying data and statistical procedures were not compromised.

JOURNAL REQUIREMENTS

Response: Thank you for your helpful suggestion. We have adjusted the heading levels to align with the required formatting guidelines and corrected the placement of asterisks in Table 1, ensuring they now appear as superscripts. Additionally, we have reviewed and updated the file names to ensure they comply with the submission requirements.

“This work was supported by the Medical Research Council and the National Institute for Health and Care Research (NIHR) [MR/V028561/1]. DA and EG received partial support of the Fogarty International Center/NIH [Training grant number D43TW00114]. DA also received partial support of the US Centers for Disease Control and Prevention cooperative agreement [Award number 5U01GH00266].”

Response: Thank you for bringing to our attention the lack of clarity in the financial disclosure. We have now included the following statement to clarify that the funders had no role in our study:

“This work was supported by the Medical Research Council and the National Institute for Health and Care Research (NIHR) [MR/V028561/1]. DA and EG received partial support of the Fogarty International Center/NIH [Training grant number D43TW00114]. DA also received partial support of the US Centers for Disease Control and Prevention cooperative agreement [Award number 5U01GH00266]. The funders had no role in study design, data collection and analysis, decision to publish, or preparation of the manuscript.”

3. We note that Figure 3 in your submission contain copyrighted images. All PLOS content is published under the Creative Commons Attribution License (CC BY 4.0), which means that the manuscript, images, and Supporting Information files will be freely available online, and any third party is permitted to access, download, copy, distribute, and use these materials in any way, even commercially, with proper attribution. For more information, see our copyright guidelines: http://journals.plos.org/plosone/s/licenses-and-copyright.

a. You may seek permission from the original copyright holder of Figure 3 to publish the content specifically under the CC BY 4.0 license.

Response: Thank you for pointing out that Figure 3 contained copyrighted images. We have removed the figure and updated the numbering of the remaining figures accordingly.

Response: Thank you for pointing out the lack of clarity regarding the source of the basemaps used in the creation of Figure 1. The basemaps were obtained from the Instituto Nacional de Estadística e Informática and the Infraestructura de Datos Espaciales del Perú, both of which provide public domain data. These can be accessed at the following websites, respectively: https://ide.inei.gob.pe/ and https://www.idep.gob.pe/.

To ensure proper attribution in accordance with the journal’s guidelines, we have added the following statement to the legend of Figure 1:

Line 102: “Fig 1. Map of the province of Huancayo. The main map presents the streets (dark grey lines) and points the studied wet markets (red dots) and laboratories (blue dots). The map of the Junin region presents the provinces of Huancayo (green) and Jauja (yellow), while the map of Peru presents the Junin region (dark grey). These maps were created using public domain basemaps provided by the “Instituto Nacional de Estadística e Informática” (https://ide.inei.gob.pe/) and the “Infraestructura de Datos Espaciales del Perú” (https://www.idep.gob.pe/).”

5. We note that there is identifying data in the Supporting Information file <S4_table.csv>. Due to the inclusion of these potentially identifying data, we have removed this file from your file inventory. Prior to sharing human research participant data, authors should consult with an ethics committee to ensure data are shared in accordance with participant consent and all applicable local laws.

-Location data

Please remove or anonymize all personal information, ensure that the data shared are in accordance with participant consent, and re-upload a fully anonymized data set. Please note that spreadsheet columns with personal information must be removed and not hidden as all hidden columns will appear in the published file.

Response: Thank you for highlighting the presence of potentially identifiable information. To ensure the privacy of study participants, we have removed the columns containing ID numbers (“id” and “clinicalrecord”). In addition, we have included the “self-perceived risk of COVID-19” variable, along with the derived variables used in our analysis, to enhance the transparency and reproducibility of our findings. The revised dataset now includes the following columns:

- Market of procedure – “market”

- Age (years) – “age”

- Age group – “age_group”

- Sex – “sex”

- Body Mass Index – “examination_imc”

- Obesity – “obesity”

- Systolic Blood Pressure – “examination_systolic”

- Diastolic Blood Pressure – “examination_diastolic”

- Hypertension – “hypertension”

- Random Blood Sugar – “examination_glucose”

- Diabetes – “diabetes”

- COVID-19 vaccination scheme – “cr_vaccination”

- COVID-19 booster – “booster”

- No self-medication practices – “selfmed_none”

- Acetylsalicylic acid consumption – “cr_selfmed_aspirin”

- Ivermectin consumption – “cr_selfmed_ivermectin”

- Chlorine dioxide consumption – “cr_selfmed_chlorinediox”

- Paracetamol consumption – “cr_selfmed_paracetamol”

- Ibuprofen consumption – “cr_selfmed_ibuprofen”

- Corticosteroids consumption – “cr_selfmed_corticosteroids”

- Azithromycin/clarithromycin consumption – “cr_selfmed_azitromicin”

- Penicillin/amoxicillin/ceftriaxone consumption – “cr_selfmed_penicilin”

- Ciprofloxacin/levofloxacin consumption – “cr_selfmed_ciprofloxacin”

- Antibiotics consumption – “selfmed_antibiotics”

- Enoxaparin consumption – “cr_selfmed_enoxaparin”

- N-acetylcysteine consumption – “cr_selfmed_nacetylcysteine”

- B complex consumption – “cr_selfmed_bcomplex”

- Zinc consumption – “cr_selfmed_zinc”

- Vitamin D consumption – “cr_selfmed_vitamind”

- Vitamin C consumption – “cr_selfmed_vitaminc”

- Other prophylactics consumption – “cr_selfmed_other”

- Self-perceived risk of COVID-19 – “cr_perceivedrisk”

- SARS-CoV-2 diagnosis – “dx_sarscov2”

- Case-control matching group – “match”

Additional Editor Comments:

To further strengthen this valuable contribution, we recommend addressing several key points:

First, regarding essential statistical validation, as noted by Reviewer 2, additional model diagnostics would enhance the findings. We recommend including appropriate goodness-of-fit measures for the conditional logistic regression and assessing potential collinearity among key predictor variables, particularly given the significant associations reported with prophylactic medications.

Response: Thank you for your valuable comments to improve our analysis. In response, we have included the calculation of the Akaike Information Criterion (AIC) for each adjusted model and complemented our model-building strategy with likelihood ratio tests, as detailed below in “Methods” and “Result” section:

In “Methods” section

Line 297: “Then, remaining variables were entered into a model without interactions following a direct approach. Subsequently, predictors with the weakest evidence of association were iteratively removed, assessing changes in estimated coefficients for remaining variables. This process was complemented by likelihood ratio tests, performed using the “lrtest” function from the “lmtest” package in R [22]. If estimated coefficients varied by 5% or more and the likelihood ratio test was significant (p-value < 0.05), the predictor was deemed contributory to the final model and reintroduced. Otherwise, it was considered non-contributory and discarded. This process continued until only predictors with strong evidence of association and significant likelihood ratio test remained. Subsequently, interaction effects between each drug and COVID-19 booster were introduced i

---

## [Decision Letter · Decision Letter 1]

Dear Dr. Guitian,

Thank you for submitting your manuscript to PLOS ONE. After careful consideration, we feel that it has merit but does not fully meet PLOS ONE’s publication criteria as it currently stands. Therefore, we invite you to submit a revised version of the manuscript that addresses the points raised during the review process.

Thank you for your thorough and thoughtful responses to the first round of reviewer and editorial comments. I am pleased to note that you have addressed the majority of the concerns raised, and the manuscript has improved as a result.

At this stage, only two minor points remain, as highlighted by Reviewer 1, which require your attention before we can proceed to acceptance:

Recruitment and Selection Bias:Please provide a more detailed description of how participants were recruited into the study. Specifically, clarify whether all eligible individuals were approached, if any were excluded or refused participation, and whether a random or consecutive sampling method was used. If not all eligible individuals were included, please report the rejection rate and provide any relevant details on the selection process. This will help readers better understand the potential for selection bias and its possible impact on your findings.Questionnaire Validation:Please specify whether the questionnaire used in your study was previously validated. If so, provide details on the validation process, including the population and setting. If the questionnaire was developed specifically for this study (ad hoc), please discuss its development process and any limitations associated with its use, such as potential measurement error or recall bias.

Once these two aspects are addressed—either by modifying the manuscript or by providing a detailed response to the reviewer—we will be able to proceed to acceptance.

Thank you again for your careful attention to the review process and for your valuable contribution to the field. We look forward to receiving your revised manuscript.

Kind regards,

Yordanis Enríquez Canto, Ph.D.

Academic Editor

PLOS ONE

Journal Requirements:

Reviewers' comments:

Reviewer's Responses to Questions

**Comments to the Author**

Reviewer #1: All comments have been addressed

Reviewer #2: All comments have been addressed

2. Is the manuscript technically sound, and do the data support the conclusions?

Reviewer #1: Yes

Reviewer #2: Yes

3. Has the statistical analysis been performed appropriately and rigorously?

Reviewer #1: Yes

Reviewer #2: Yes

4. Have the authors made all data underlying the findings in their manuscript fully available?

Reviewer #1: Yes

Reviewer #2: Yes

5. Is the manuscript presented in an intelligible fashion and written in standard English?

Reviewer #1: Yes

Reviewer #2: Yes

Reviewer #1: The manuscript addresses the previously raised comments; however, I have a few minor suggestions for further improvement:

1. Recruitment and Selection Bias:

The manuscript should clearly explain how participants were recruited into the program. This aspect, while noted in the limitations, may represent a potential source of selection bias, and a more detailed description would help the reader better understand the possible direction and magnitude of this bias.

Was a random sampling method employed, or were all eligible individuals included? If not all were included, please report the rejection rate and provide any relevant details on the selection process.

2. Questionnaire Validation:

It is essential to specify whether the questionnaire used in the study has been previously validated. If so, please provide details on the validation process, including the population and setting in which it was validated. If it is an ad hoc instrument, a discussion on its development and limitations is necessary.

Reviewer #2: The statistical procedure was duly considered and clarified. The results were explained, and collinearity was assessed using Cramér’s V test.

**Do you want your identity to be public for this peer review?** For information about this choice, including consent withdrawal, please see our Privacy Policy

Reviewer #1: No

Reviewer #2: No

---

## [Author Response · Author response to Decision Letter 2]

17 Jun 2025

Dear Editor,

We sincerely thank the reviewers and the editorial team for their thoughtful comments and suggestions, which have substantially contributed to improving our manuscript (PONE-D-25-09334R1). In response, we have carefully revised the manuscript and are pleased to submit the updated version for your consideration.

All comments have been addressed in the accompanying document. Our responses are presented in green, and the corresponding edits to the manuscript are shown in red.

REVIEWERS’ COMMENTS

REVIEWER #1

The manuscript addresses the previously raised comments; however, I have a few minor suggestions for further improvement:

1. Recruitment and Selection Bias: The manuscript should clearly explain how participants were recruited into the program. This aspect, while noted in the limitations, may represent a potential source of selection bias, and a more detailed description would help the reader better understand the possible direction and magnitude of this bias.

Was a random sampling method employed, or were all eligible individuals included? If not all were included, please report the rejection rate and provide any relevant details on the selection process.

Answer: We appreciate the suggestion to further clarify the recruitment process employed in our study. In response, we have included additional details in the Methods section to provide a more comprehensive description of how participants were recruited. The revised text reads as follows:

Line 110: “The program comprised of three sequential phases conducted before, during, and after the third wave of SARS-CoV-2 infections. During the first phase, from August to December 2021, the program worked with market vendors to establish its objectives, and consultation rooms were set up in locations within each market as designated by the vendors. Also, health workers assigned to each market oversaw participant registration and monitoring throughout the program. Recruitment was conducted through direct communication from market authorities to all workers in each market. Participation was voluntary and based on convenience sampling, as individuals self-enrolled by presenting themselves at the consultation room from their respective markets. No formal rejections were recorded, although not all eligible workers chose to participate. Finally, registration involved an interviewer-administered questionnaire, a clinical examination, and a nasopharyngeal swab for an antigen SARS-CoV-2 test (Roche Diagnostics SL, Barcelona, Spain), along with voluntary validation of COVID-19 vaccination status using individual vaccination cards given by the Peruvian Ministry of Health.”

Finally, we have also addressed the potential selection bias associated with the recruitment approach used in this study by revising and reorganizing the first paragraph of the limitations as follows:

Line 549: “Our study has limitations. First, the results found in our study may not be fully generalizable, as participants were exclusively traditional market vendors, and recruitment was conducted through a non-random, convenience-based approach. Specifically, participation was voluntary and limited to individuals who actively approached the registration office in response to announcements made by market authorities. This self-selection process may have led to selection bias, as those more interested or concerned about COVID-19 might have been more likely to enroll. In addition, no formal records of refusals during the first phase of the health program, further limiting the ability to assess the direction or magnitude of this potential bias. Nonetheless, according to the last National Census of Food Markets in 2016 [48], Market A reported approximately 600 operational stalls and Market B reported 1,000. In comparison, our study initially recruited 224 vendors from Market A and 215 from Market B, suggesting that a substantial fraction of the total vendor population was recruited although the obtained sample could be potentially biased. Further, traditional markets could still serve as potential SARS-CoV-2 sentinel surveillance sites, given their constant interaction with the general population.”

In order to address this suggestion adequately, we have added the following reference in the final version of this manuscript:

48. Instituto Nacional de Estadística e Informática. Censo Nacional de Mercados de Abasto. 2016: Resultados a nivel nacional [National Census of Food Markets 2016: National-level results]. Lima, Peru: Instituto Nacional de Estadística e Informática (INEI); 2017. Available: https://www.inei.gob.pe/media/MenuRecursivo/publicaciones_digitales/Est/Lib1448/libro.pdf

2. Questionnaire Validation: It is essential to specify whether the questionnaire used in the study has been previously validated. If so, please provide details on the validation process, including the population and setting in which it was validated. If it is an ad hoc instrument, a discussion on its development and limitations is necessary.

Answer: Thank you for highlighting the need for greater clarity regarding the validation process of the questionnaire used in our study. We have now specified that it was an ad hoc instrument, developed specifically for the purposes of this research. The validation process involved a content evaluation conducted by a panel of subject-matter experts and a pilot study conducted in an external market, as follows:

Line 207: “A structured interviewer-administered questionnaire was employed to collect data on participant’s characteristics. The questionnaire was developed by the authors after a detailed review of the literature. As it was an ad hoc instrument, its content validity was evaluated by a panel of subject-matter experts appointed during the ethical review process. The experts assessed the questionnaire for clarity, relevance, and alignment with the study objectives. Minor revisions were made based on their feedback. Finally, a pilot test was conducted with vendors from an external traditional market to evaluate participants’ comprehension and ensure the questions were appropriately understood. Additional minor adjustments were made based on insights gained from the pilot study to enhance clarity and consistency.”

Also, we have discussed on its development and limitations as follows:

Line 595: “Sixth, the questionnaire used was an ad hoc instrument developed specifically for the purposes of this study. Although it underwent expert review as part of the ethical assessment process, and was piloted in a group of vendors from an external traditional market to evaluate clarity and participant understanding, the questionnaire may still present limitations in its psychometric robustness and could introduce measurement bias in self-reported behaviours.”

REVIEWER #2

The statistical procedure was duly considered and clarified. The results were explained, and collinearity was assessed using Cramér’s V test.

Answer: We sincerely appreciate the valuable suggestions provided during the previous revision round, which have significantly contributed to enhancing the quality of this manuscript.

Best regards,

Javier Guitian

---

## [Decision Letter · Decision Letter 2]

Association between self-administrated prophylactics and SARS-CoV-2 infection among traditional market vendors from the Central Highlands of Peru: A nested case-control study

PONE-D-25-09334R2

Dear Dr. Guitian,

We’re pleased to inform you that your manuscript has been judged scientifically suitable for publication and will be formally accepted for publication once it meets all outstanding technical requirements.

Kind regards,

Yordanis Enríquez Canto, Ph.D.

Academic Editor

PLOS ONE

Additional Editor Comments (optional):

Reviewers' comments:

Reviewer's Responses to Questions

**Comments to the Author**

Reviewer #1: All comments have been addressed

2. Is the manuscript technically sound, and do the data support the conclusions?

Reviewer #1: Yes

3. Has the statistical analysis been performed appropriately and rigorously?

Reviewer #1: Yes

4. Have the authors made all data underlying the findings in their manuscript fully available?

Reviewer #1: Yes

5. Is the manuscript presented in an intelligible fashion and written in standard English?

Reviewer #1: Yes

Reviewer #1: (No Response)

**Do you want your identity to be public for this peer review?** For information about this choice, including consent withdrawal, please see our Privacy Policy

Reviewer #1: **Yes: ** Wendy Nieto-Gutierrez

---

## [Editor Report · Acceptance letter]

PONE-D-25-09334R2

PLOS ONE

Dear Dr. Guitian,

I'm pleased to inform you that your manuscript has been deemed suitable for publication in PLOS ONE. Congratulations! Your manuscript is now being handed over to our production team.

Kind regards,

on behalf of

Prof. Yordanis Enríquez Canto

Academic Editor

PLOS ONE